# THE ROLE OF ACTIVE LEARNING IN MODERN DEEP LEARNING

## ABSTRACT

Even though Active Learning (AL) is widely studied, it is rarely applied in contexts outside its own scientific literature. We posit that the reason for this is AL's high computational cost coupled with the comparatively small lifts it is typically able to generate in scenarios with few labeled points. In this work we study the practical setup of exhausting a fixed budget to label points from a large unlabeled pool and designing a training pipeline to train the strongest possible model on this small labeled set. We compare the impact of different methods to combat this low data scenario, namely data augmentation (DA), semi-supervised learning (SSL) as options for the training pipeline and AL as selection strategy for the labeled points. We find that AL is by far the least efficient method of solving the low data problem, generating a lift of only 1-4% over random sampling, while DA and SSL methods can generate up to 60% lift in combination with random sampling. However, when AL is combined with strong DA and SSL techniques, it surprisingly is still able to provide improvements. Based on these results, we frame AL not as a method to combat missing labels, but as the final building block to squeeze the last bits of performance out of data after appropriate DA and SSL methods as been applied.

## 1 INTRODUCTION

Training ML models in most real use cases entails working with limited amounts of labeled data. Since labels are expensive to obtain, datasets usually are split into a small labeled pool and a much larger unlabeled pool. To obtain the labeled pool, a fixed budget is allocated to acquire labels. The method to select which points should be labeled can be varied between random sampling and more sophisticated acquisition functions. The training pipeline to obtain a model from the labeled set may also vary in the employed supporting features besides the loss function.

In this paper we focus on classification problems and provide insights about the three most researched techniques to train strong models under these constraints: data augmentation (DA), semi-supervised learning (SSL) and active learning (AL). Even though all three techniques work differently (DA increases the amount of labeled data, SSL makes use of unlabeled data and AL tries to improve the selection of points that are labeled), all of them solve the same problem of limited availability of labeled data. In this sense, AL directly competes with DA and SSL as strategies for enhancing model quality in low-label regimes. Current literature has yet provided a comprehensive study of the combined application of all three methods, researching the question of which method works best in isolation, as well as whether they can be freely combined (with each consecutive method still providing a lift). In this work, we employ two well known DA methods and a collection of well-performing AL algorithms from a recent benchmark (Werner et al., 2024). As SSL paradigm, we chose pretraining as the most used SSL paradigm in recent literature.

We pay special attention to the performance of active learning methods as it changes with different training pipelines, as techniques like DA or SSL are rarely used in AL literature. As a motivating example, we are comparing random sampling with various advanced training protocols containing DA and/or SSL against the **best** performing AL algorithm without DA or SSL on that dataset. From Fig. 1 you can observe that active learning techniques fall behind DA or SSL methods in terms of how much lift they provide over a randomly sampled labeled set and plain supervised learning without augmentations.

This elicits the question, whether AL is a useful technique to combat low data scenarios at all, or if

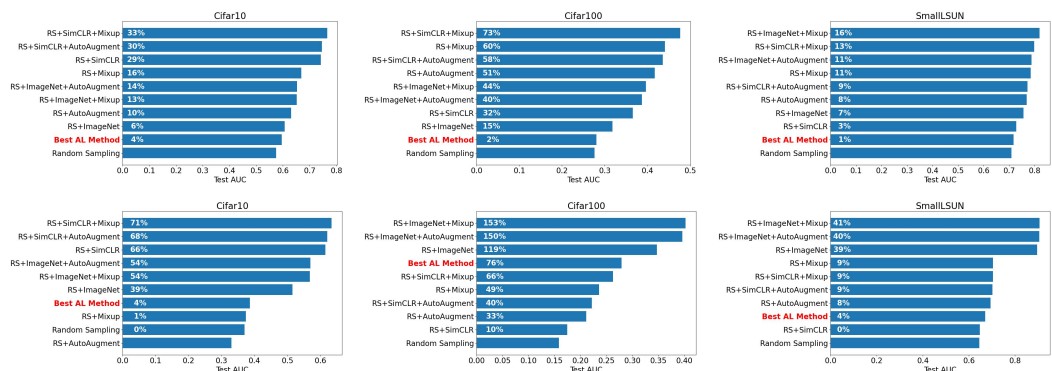

Figure 1: Performance of Random Sampling (RS) plus different DA and SSL methods and the best AL method without DA or SSL. Numbers in each bar indicate the percentage improvement over random sampling. Top row: ResNet18, bottom row: EfficientNetV2

DA and SSL already exhaust the available lifts. To the best of our knowledge, no methodological paper in AL has tested their proposed algorithm in a regime with strong DA and SSL methods and only one benchmark paper (Lüth et al., 2024) provides tertiary experiments about the combination of all three methods. Considering the high computational cost of AL, a systematic study on the impact of AL on a modern training pipeline including DA and SSL provides a valuable answer to the question "Do we need AL in modern machine learning?". To this end, we study different combinations of DA and SSL techniques for three different datasets and test, whether AL methods can provide an additional lift on the optimal setup per dataset. [1]

KEY INSIGHTS

1. Active Learning is the least efficient method of overcoming low data scenarios

2. Despite that, active learning can still provide lifts even when paired with strong data augmentation and semi-supervised learning techniques

3. In this regime, every competitive active learning method performs exactly on par

## 2 PROBLEM DESCRIPTION

We are experimenting on pool-based AL with classification models. Mathematically we have the following:

Given a dataset $\mathcal{D}_{\text{train}} := \{(x_i, y_i) \quad i \in \{1, \ldots, N\}\}$ with $x \in \mathcal{X}, y \in \mathcal{Y}$ (following (Werner et al., 2024) we similarly have $\mathcal{D}_{\text{val}} := \{(x_i, y_i) \quad i \in \{1, \ldots, N_{\text{val}}\}\}$ and $\mathcal{D}_{\text{test}} := \{(x_i, y_i) \quad i \in \{1, \ldots, N_{\text{test}}\}\})$ we randomly sample an initial labeled pool $L^{(0)} \sim \mathcal{D}_{\text{train}}$ that we call the seed set. We suppress the labels from the remaining samples to form the initial unlabeled pool $U^{(0)} = \mathcal{D}_{\text{train}}/L^{(0)}$. We define an acquisition function to be a function that selects a batch of samples of size $\tau$ from the unlabeled pool $a(U^{(i)}) := \{x_b^{(i)}\} \in U^{(i)} \quad b := [0, \ldots, \tau]$. We then recover the corresponding labels $y_b^{(i)}$ for these samples and add them to the labeled pool $L^{(i+1)} := L^{(i)} \cup \{(x_b^{(i)}, y_b^{(i)})\}$ and $U^{(i+1)} := U^{(i)}/\{x_b^{(i)}\} \quad b := [0, \ldots, \tau]$. The acquisition function is applied until a budget $B$ is exhausted.

We measure the performance of a model $\hat{y} : \mathcal{X} \to \mathcal{Y}$ on the held out test set $\mathcal{D}_{\text{test}}$ after each acquisition round by fitting the model $\hat{y}^{(i)}$ on $L^{(i)}$ and measuring the test accuracy.

We allow the fitting process to additionally depend on a DA technique and an SSL technique who aid model training: $\text{TRAIN}(\hat{y}^{(i)}, L^{(i)}, \text{DA}, \text{SSL})$. DA is allowed to alter the labeled samples of that iteration: $\text{DA}(L^{(i)})$, while SSL can make use of either a fully labeled external dataset $\mathcal{D}_{\text{ext}} \cap \mathcal{D}_{\text{train}} = \emptyset$ and/or the unlabeled pool of our current dataset: $\text{SSL}(\hat{y}^{(i)}, \{\mathcal{D}_{\text{ext}}, U^{(i)}\})$.

---

[1] Code available. Sec. 8 "Reproducibility" for details.

## 3 RELATED WORK

We are currently aware of only one methodological paper (Chan et al., 2021) that combines strong supporting techniques with their proposed method. The authors test Coreset (Sener & Savarese, 2017) and VAAL (Sinha et al., 2019) in a framework with Debiased Contrastive Learning (DCL) and FixMatch and find no further lift from AL. This is consistent with our results that diversity-based methods do not work in this setting (see Sec. 6 for details). We extend the study of (Chan et al., 2021) by testing different set of SSL techniques, adding DA and greatly extending the range of tested AL methods.

Some recent benchmark papers have also studied aspects of DA and SSL: (Beck et al., 2021) found that DA does not only improve the overall test accuracy of BADGE (Ash et al., 2020), but also its label efficiency. (Werner et al., 2024) test AL methods in an SSL setting by pre-encoding their datasets with a pretrained encoder, but do not employ DA in any of their experiments. (Lüth et al., 2024) propose to tune DA as part of the hyperparameters in the first iteration of AL, as well as evaluating AL in two SSL scenarios. We extend the study of (Lüth et al., 2024) in three ways: First, by allowing a comprehensive evaluation of DA and SSL techniques on the tested datasets in order to find the optimal combination, second, by quantifying how much each method contributes to overcoming the low data problem, and third, by significantly extending the list of tested AL algorithms.

Additional related work comes from (Tamkin et al., 2022), who study the behavior of actively learned pretrained models under domain shift and synthetic datasets with spurious features and find uncertainty sampling to be beneficial in these domains.

## 4 METHODOLOGY

This work serves as a guide for machine learning practitioners tasked with training high-performing models on unlabeled datasets. In many cases, random sampling combined with simple techniques like DA and leveraging pretrained ImageNet weights are the only methods applied, due to their low computational cost and accessibility. We study the impact of various DA and semi-supervised learning (SSL) techniques when used alongside random data selection, and explore whether active learning (AL) can provide additional improvements in these settings.

We argue that the effectiveness of an AL method is not necessarily independent of the presence of DA or SSL. Marginal improvements from AL (as seen in Fig. 1) may be overshadowed by stronger techniques. To address this, we propose a series of three experiments: (i) Analyzing the individual ability of DA, SSL and AL of improving upon the random sampling baseline with vanilla supervised training, (ii) finding the optimal combination of DA and SSL for each dataset and (iii) testing whether AL is able to provide a lift over random sampling combined with optimal DA and SSL.

To evaluate our experiments, we measure test accuracy of our classifier in $i$ rounds, where each classifier $\hat{y}^{(i)}$ is trained on $L^{(i)}$ with $i \in [1 \ldots B/\tau]$. Each round we add 500 samples to our labeled pool ($\tau = 500$). As aggregate metric we are using the normalized area under the accuracy curve (AUC):

$$\text{AUC}(\mathcal{D}_{\text{test}}, \hat{y}, B) := \frac{1}{B/\tau} \sum_{i=1}^{B/\tau} \text{Acc}(\mathcal{D}_{\text{test}}, \hat{y}^{(i)}) \tag{1}$$

A higher AUC signifies better average performance across $i$ iterations of measuring. Note that this protocol is also followed for experiments using only random sampling in combination with DA or SSL. Even though we could randomly sample $B$ points all at once and train a single model, we opt for a unified protocol and the use of AUC values for two reasons: (i) AUC is the preferred method of evaluating iterative AL algorithms like BADGE and, this way, we obtain directly comparable results and (ii) the AUC is less dependent on the chosen budget. A comparison based on the final accuracy for any high budget might be meaningless for lower budgets of practical applications. The AUC incorporates this information in its score. Furthermore, we repeat every experiment 20 times and compare the results with paired-t-tests and Critical Difference diagrams, adhering to the best practices proposed by (Werner et al., 2024). Additionally, we report the learning trajectories of all tested methods in App. C. The investigated methods are AutoAugment (Cubuk et al., 2019) and Mixup (Zhang et al., 2017) for DA and pretrained ImageNet weights and SimCLR (Chen et al., 2020) for SSL. For AL, we incorporate all well-performing AL methods from (Werner et al., 2024),

| | Cifar10 | Cifar100 | SmallLSUN |
|---|---|---|---|
| Query Size | 500 | 500 | 500 |
| Budget | 10k | 30k | 20k |
| #Classes | 10 | 100 | 6 |
| Imgs per Class | 6000 | 600 | 10k |
| Img Size | 32 | 32 | 224 |

Table 1: Statistics of the employed datasets. SmallLSUN is composed of 6 classes from the Large-scale Scene Understanding (LSUN) dataset (Wang et al., 2017). For details, please refer to Appendix A.

namely Badge (Ash et al., 2020), Galaxy (Zhang et al., 2022), Uncertainty Sampling (Entropy, Margin, Least Confident), Coreset (Sener & Savarese, 2017) and CoreGCN (Caramalau et al., 2021). For a summary of employed datasets and their chosen budgets, refer to Table 1. Please note, that we can not use any dataset that is derived from ImageNet, as we are using ImageNet-weights as a pretraining option for our classifers.

## 5 IMPLEMENTATION DETAILS

In this work, we are using ResNet18 and EfficientNetV2 as our classification models since both architectures have proven to be very stable for a variety of datasets. We deliberately choose not to optimize our hyperparameters in a search, but rather use the default settings of our chosen optimizer.

This is on one hand adhering to the validation paradox described in (Lüth et al., 2024), where an optimal set of hyperparameters for AL cannot be found as this would entail labeling excessive amounts of extra data, on the other hand we argue that uncertainty sampling methods profit unproportionally from optimized hyperparameters. Evidence of this can be seen in the recent benchmark paper of (Werner et al., 2024), where Least-Confidence Sampling was the best performing AL method in the vision domain, and Margin Sampling being the best method over all. In order to enable a fair comparison between uncertainty- and diversity-based AL methods, we use default choices for our hyperparameters. For further details of our employed hyperparameters, please refer to Appendix B.

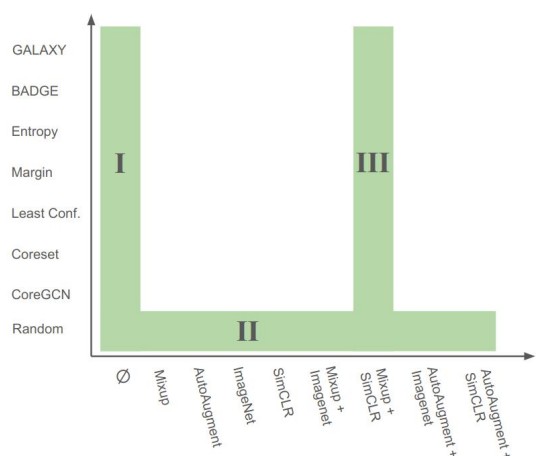

Figure 2: Overview of conducted experiments on Cifar100 with a ResNet18. Green areas indicate tested combinations. Each entry on the x-axis uses random sampling.

Our SimCLR pretraining is identical to (Werner et al., 2024) with 100-200 epochs of unsupervised contrastive training with SGD and a cosine learning rate scheduler.

Every experiment is repeated 20 times with different random seeds.

## 6 EXPERIMENTS

An exemplary overview of our conducted experiments can be found in Fig. 2. Areas shaded in green indicate tested combinations, while other regions have been omitted due to prohibitive computational costs. First, we measure individual lifts of DA, SSL and AL methods (area I and II) by comparing their AUC values to the baseline AUC of random sampling with vanilla supervised training. We select the best performing AL method without DA/SSL as the representative for AL for Fig. 1 and our experiments described below. The number in each bar indicates the percentage lift over the baseline. Individual performances of all AL methods on all datasets and model architectures can be found in App. C.

We observe that AL methods significantly lag behind all other methods to combat low data scenarios, with the sole exception of EfficientNetV2 on Cifar100, where AL is still only 50% as effective as ImageNet + Mixup. This is especially impactful, considering the computational cost of the average AL setup. Since AL requires training of $B/\tau$ many classifiers, it is roughly $B/\tau$ times more expensive than any DA technique or ImageNet weights. A direct comparison to the computational cost of SimCLR challenging, as the pretraining time varies between datasets and model architectures. In our case, the SimCLR pretraining took longer than a single run of e.g. BADGE sampling, but this might change depending on chosen hyperparameters. At the same time, SimCLR is generally offering a greater lift than any tested AL method. From this experiment we conclude that AL alone is not efficient in overcoming the low data scenario, as much cheaper techniques with greater lift should always be employed instead.

Our second observation from Fig 1 is that DA and SSL techniques generally stack well, i.e. they do not overshadow each other's lifts. This is no novel insight, as many modern training pipelines for vision datasets successfully include both DA and SSL techniques. AL literature, however, is rarely incorporating either, let alone both, eliciting the question, whether the lifts of AL methods also stack similarly. To this end, we tested our AL methods on the best performing combination of DA and SSL for each dataset (area III in Fig. 2) creating the hardest possible environment to produce further lifts in. We display the results for LSUN (EfficientNetV2) in Fig. 3 and Cifar100 (ResNet18) in Fig. 4. Remaining results can be found in App. C.

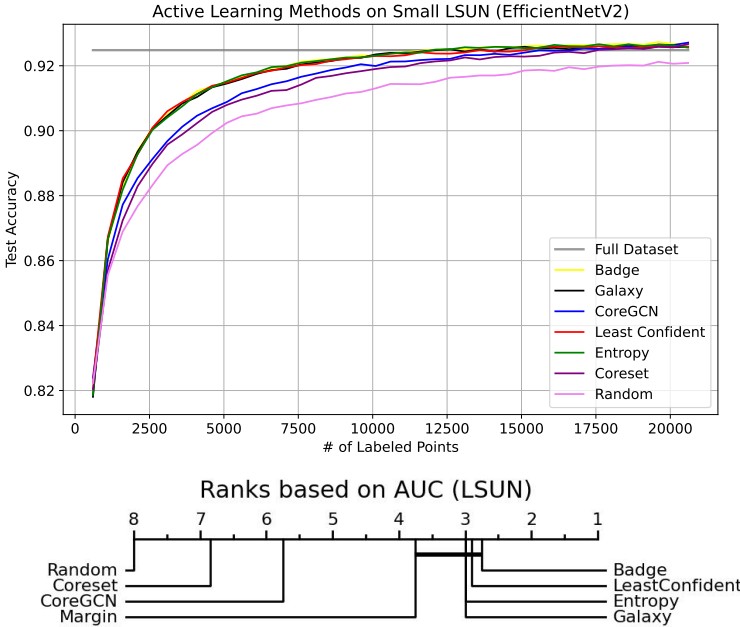

Figure 3: Test accuracy curves for all AL methods for LSUN on the best performing DA/SSL setup with EfficientNetV2 (top) and resulting CD-Diagram (bottom)

Finally, we display the aggregated performance of all AL methods across all datasets and model architectures in Fig. 5. For each combination, the optimal combination of DA/SSL was used to create the hardest possible environment for AL to produce lifts in. From Fig. 5 we observe that, on average, all AL methods are still able to perform better than random sampling. However, historically strong AL methods that rely on uncertainty sampling outperform diversity based methods by a large margin. Additionally, the gap between well-performing methods is no longer statistically significant, as indicated by the bar between "Margin" and "Badge" in Fig. 5. This cluster of well-performing methods can also be observed in the learning trajectories of Fig. 3 and 4, where all methods achieve the same final test accuracy with only minor differences in the early iterations. Even though the critical difference diagrams indicate an advantage of Badge in Fig 5, the difference in test accuracy

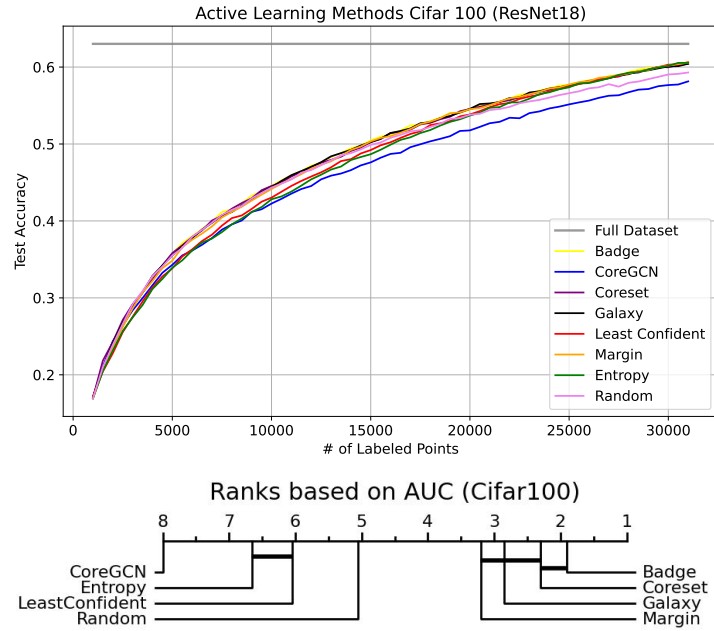

Figure 4: Test accuracy curves for all AL methods for Cifar100 on the best performing DA/SSL setup with ResNet18 (top) and resulting CD-Diagram (bottom)

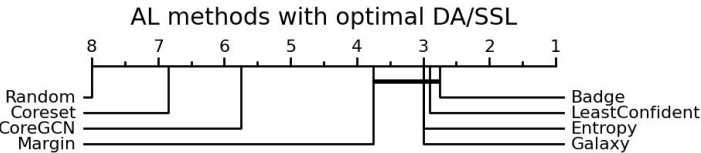

Figure 5: Ranks of AL methods across every dataset and model architecture. For each combination, the optimal combination on DA/SSL was used.

is marginal. Accuracy curves for all combinations of dataset and model architecture are qualitatively the same and can be found in App C.

## 7 CONCLUSION

In this work we quantified the individual impact of DA, SSL and AL methods on small randomly sampled, labeled pools. We found that AL is by far the least efficient method of improving upon the low data scenario, since it offers only a 1-4% lift (except EfficientNetV2 on Cifar100) over random sampling with a significant investment of compute. A practitioner of machine learning can always employ appropriate DA and SSL techniques first and expect a better return.

After testing all AL methods with the best DA and SSL setup for each dataset, we found that while some methods improved performance, none had a significant **best** performance on any dataset. The only consistent algorithms were Badge, Galaxy and Margin sampling; so after applying DA and SSL, a practitioner is free to choose among these. This is consistent with (Lüth et al., 2024), who also found that the field of AL methods moves closer together in the presence of DA and SSL and some methods like Coreset start to collapse to random or sub-random performance.

This work serves as a guide for practitioners to design their training pipelines in a zero-shot manner: While DA and SSL are universally beneficial and oftentimes cheap to obtain (even SimCLR only has to be done once), they can decide to include AL based on the required performance on the dataset. Only if they need to obtain the final few percentage points of the possible performance on

this dataset, they should opt for AL.

We would like to close with a proposed paradigm shift in AL research: Developing an AL method in an environment without DA and SSL techniques is **not** scientifically sound, as it might interact with these techniques in unpredictable ways. Modern AL research needs to make sure that their proposed method does not collapse to random performance in modern training pipelines, and it should strive to outperform the cluster of strong uncertainty sampling methods that have been identified by this work and recent benchmarks (Werner et al., 2024; Lüth et al., 2024; Ji et al., 2023) in environments with DA and SSL.

## 8 REPRODUCIBILITY

Our entire source code is available under `anonymous.4open.science/r/The-Role-of-Active-Learning-in-Modern-Deep-Learning-4FDD`. We adhere to the reproducibility guidelines of (Werner et al., 2024) by using their proposed experimental framework as well as repeating every experiment 20 times to reduce the inter-experiment variance. All experiments have been conducted on a mix of NVIDIA 2080, 3090 and A4000 GPUs for a total computation time of $\sim 40$ GPU months.

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

## A  SMALL LSUN DATA

SmallLSUN is composed of 6 classes from the Large-scale Scene Understanding (LSUN) dataset (Wang et al., 2017). The dataset contains $224 \times 224$ images of indoor and outdoor scenes with labels referring to the location of the scene. We have compiled a list of classes from this dataset that have more than 10k examples, but remain under 16 GB of data to limit the computational burden of testing AL algorithms.

We selected the following classes from `https://www.tensorflow.org/datasets/catalog/lsun`:

1. Bridge

2. Church outdoor

3. Conference Room

4. Dining Room

5. Restaurant

6. Tower

Sampled 10k images from these 6 classes, resulting in a dataset of size 60k (train), 1.8k (validation) and 6k (test).

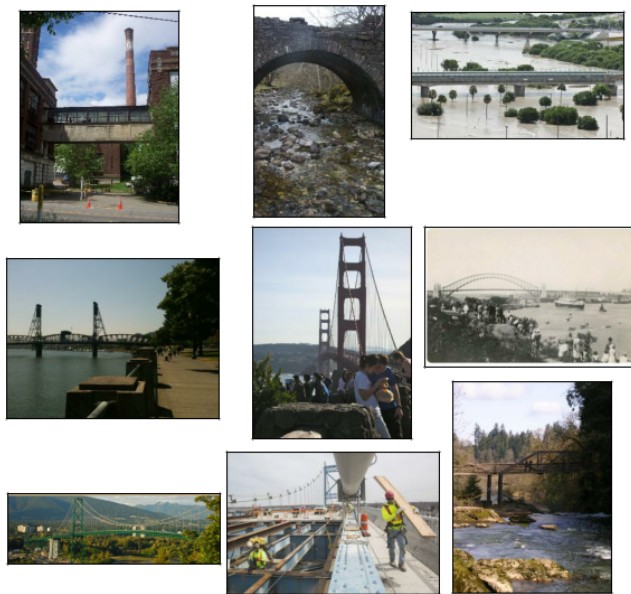

Figure 6: Example images for the Bridge class. Taken from https://www.tensorflow.org/datasets/catalog/lsun

## B  HYPERPARAMETERS

**ResNet18**

| | Cifar10 | Cifar100 | LSUN |
|---|---|---|---|
| **Evaluation** | | | |
| Optimizer | NAdam | NAdam | NAdam |
| Learning Rate | 0.001 | 0.001 | 0.001 |
| Weight Decay | 0 | 0 | 0 |
| **SimCLR** | | | |
| Epochs | 100 | 100 | 250 |
| Optimizer | SGD | SGD | SGD |
| Initial LR | 0.4 | 0.4 | 0.4 |
| LR Scheduler | Cosine | Cosine | Cosine |
| Weight Decay | 0.0001 | 0.0001 | 0.0001 |
| **Mixup** | | | |
| Probability | 0.5 | 0.5 | 0.5 |
| $\alpha$ | 1 | 1 | 1 |
| **AutoAugment** | | | |
| Probability | 0.5 | 0.5 | 0.5 |
| Policy* | Cifar10 | Cifar10 | Imagenet |

Table 2: Hyperparameters for ResNet18. "Evaluation" refers to the procedure described in Section 4. (*)AutoAugment policies taken from the PyTorch library.

**EfficientNetV2**

| | Cifar10 | Cifar100 | LSUN |
|---|---|---|---|
| **Evaluation** | | | |
| Optimizer | SGD | SGD | SGD |
| Learning Rate | 0.005 | 0.005 | 0.005 |
| Weight Decay | 5e-5 | 5e-5 | 5e-5 |
| Momentum | 0.5 | 0.5 | 0.5 |
| **SimCLR** | | | |
| Epochs | 100 | 100 | 250 |
| Optimizer | SGD | SGD | SGD |
| Initial LR | 0.25 | 0.15 | 0.2 |
| LR Scheduler | Cosine | Cosine | Cosine |
| Weight Decay | 0.0001 | 0.0001 | 0.0001 |
| **Mixup** | | | |
| Probability | 0.5 | 0.5 | 0.5 |
| $\alpha$ | 1 | 1 | 1 |
| **AutoAugment** | | | |
| Probability | 0.5 | 0.5 | 0.5 |
| Policy* | Cifar10 | Cifar10 | Imagenet |

Table 3: Hyperparameters for EfficientNetV2. "Evaluation" refers to the procedure described in Section 4. (*)AutoAugment policies taken from the PyTorch library.

# C ALL RESULTS

## CIFAR10 - RESNET18

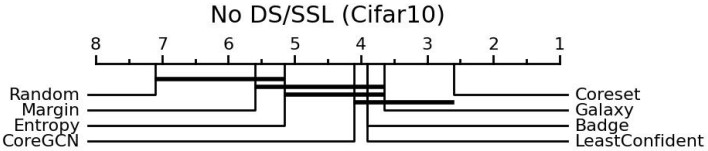

Figure 7: Ranking of AL methods on Cifar10 without DA or SSL

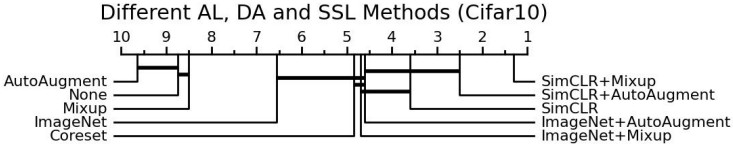

Figure 8: Ranking of **best** AL and different DA/SSL methods on Cifar10

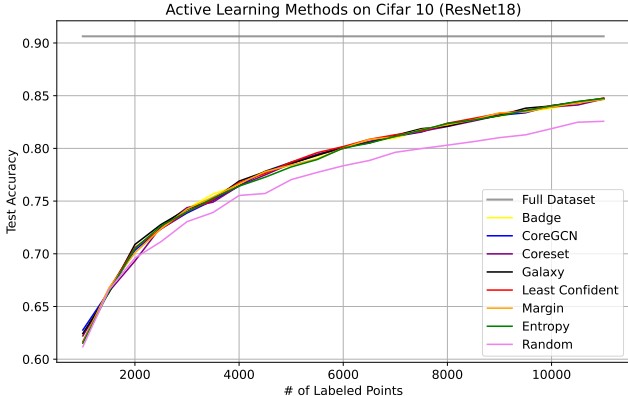

Figure 9: Test accuracy curves for Cifar10 with optimal combination of DA and SSL methods.

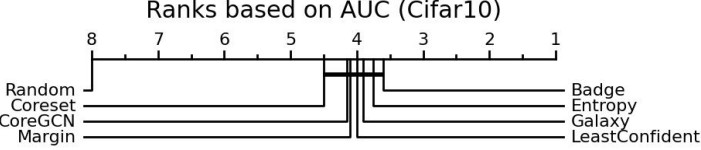

Figure 10: Ranking of AL methods on Cifar10 with optimal combination of DA and SSL methods.

CIFAR100 - RESNET18

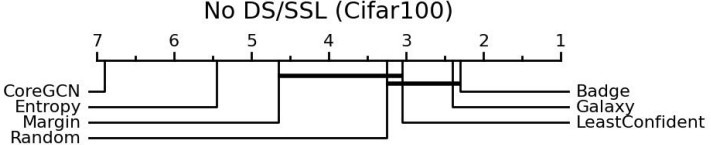

Figure 11: Ranking of AL methods on Cifar100 without DA or SSL

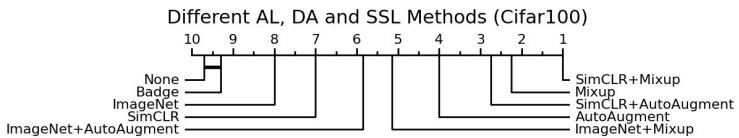

Figure 12: Ranking of **best** AL and different DA/SSL methods on Cifar100

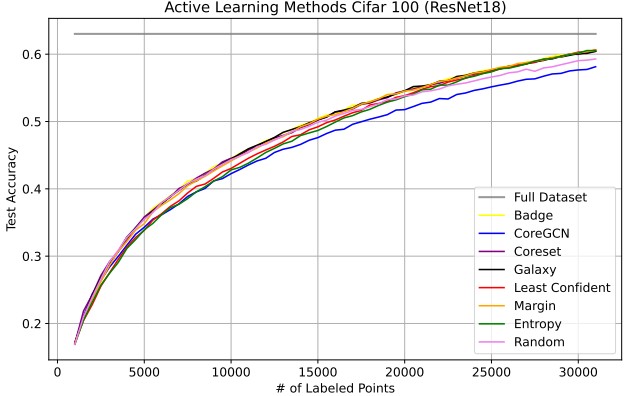

Figure 13: Test accuracy curves for Cifar100 with optimal combination of DA and SSL methods.

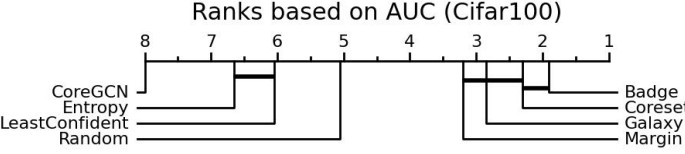

Figure 14: Ranking of AL methods on Cifar100 with optimal combination of DA and SSL methods.

LSUN - RESNET18

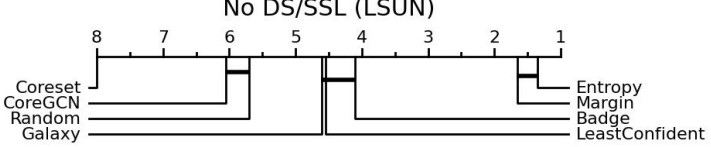

Figure 15: Ranking of AL methods on LSUN without DA or SSL

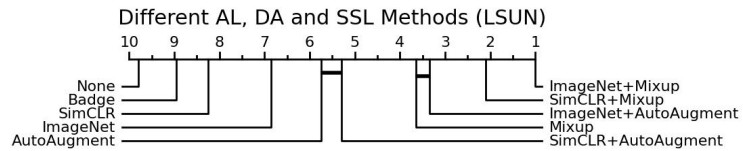

Figure 16: Ranking of **best** AL and different DA/SSL methods on LSUN

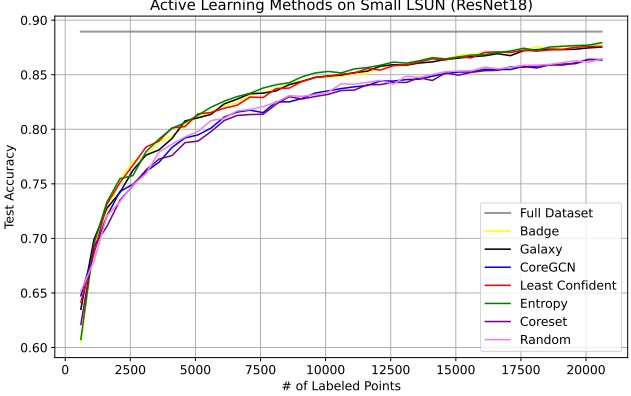

Figure 17: Test accuracy curves for LSUN with optimal combination of DA and SSL methods.

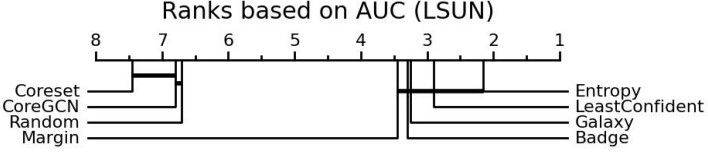

Figure 18: Ranking of AL methods on LSUN with optimal combination of DA and SSL methods.

CIFAR10 - EFFICIENTNETV2

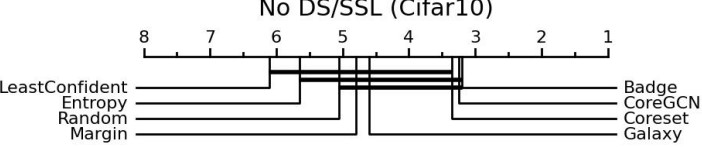

Figure 19: Ranking of AL methods on Cifar10 without DA or SSL

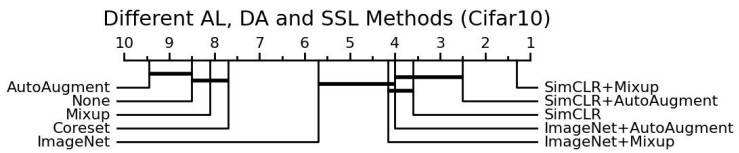

Figure 20: Ranking of **best** AL and different DA/SSL methods on Cifar10

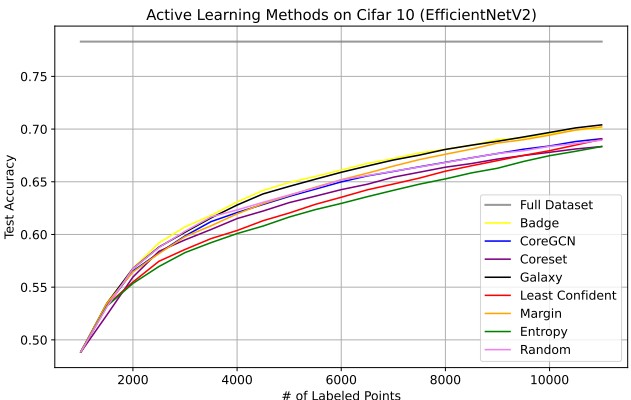

Figure 21: Test accuracy curves for Cifar10 with optimal combination of DA and SSL methods.

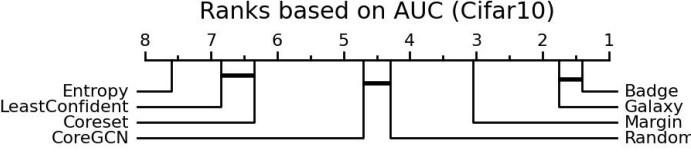

Figure 22: Ranking of AL methods on Cifar10 with optimal combination of DA and SSL methods.

CIFAR100 - EFFICIENTNETV2

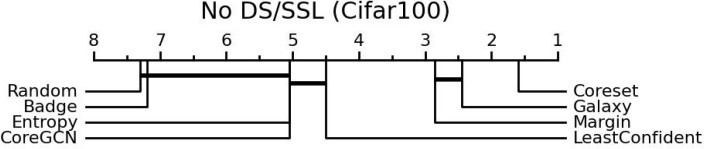

Figure 23: Ranking of AL methods on Cifar100 without DA or SSL

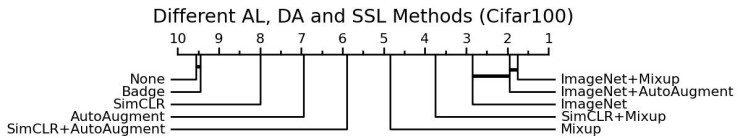

Figure 24: Ranking of **best** AL and different DA/SSL methods on Cifar100

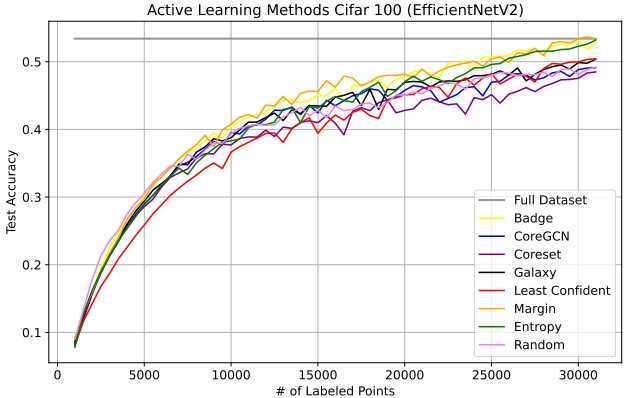

Figure 25: Test accuracy curves for Cifar100 with optimal combination of DA and SSL methods.

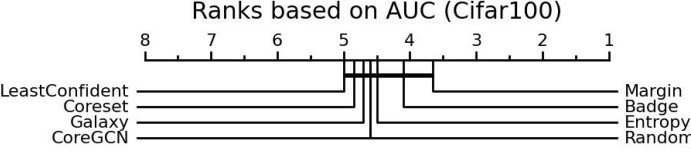

Figure 26: Ranking of AL methods on Cifar100 with optimal combination of DA and SSL methods.

LSUN - EFFICIENTNETV2

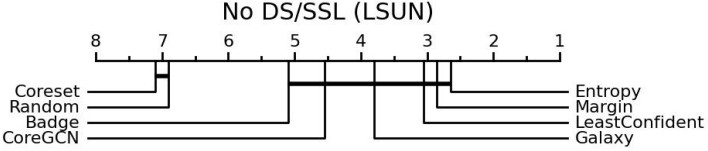

Figure 27: Ranking of AL methods on LSUN without DA or SSL

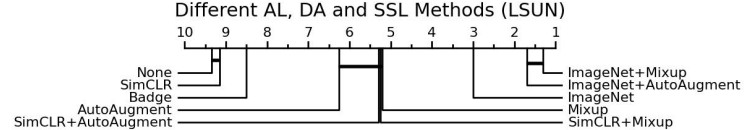

Figure 28: Ranking of **best** AL and different DA/SSL methods on LSUN

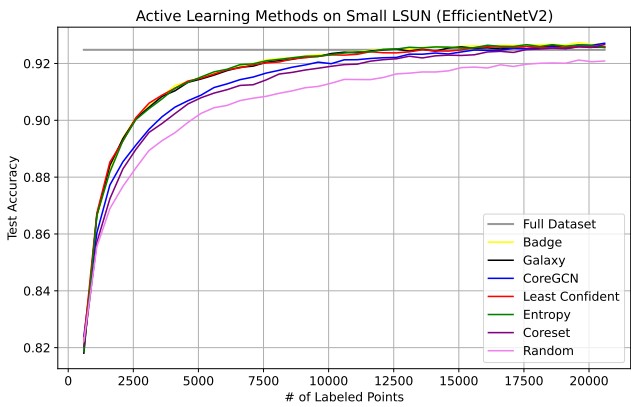

Figure 29: Test accuracy curves for LSUN with optimal combination of DA and SSL methods.

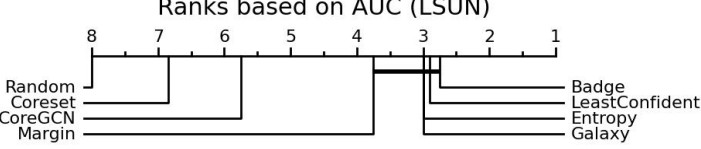

Figure 30: Ranking of AL methods on LSUN with optimal combination of DA and SSL methods.

