# OpenReview forum: "The Role of Active Learning in Modern Deep Learning"
_ICLR.cc/2026/Conference — Submitted to ICLR 2026_

### Official Review · Reviewer_WQjT · 2025-10-31

**Soundness:** 2
**Presentation:** 1
**Contribution:** 1
**Rating:** 2
**Confidence:** 4

**Summary:**

This paper investigates the combination of Active Learning with Data Augmentation and Semi-Supervise Learning in classification tasks (CIFAR10, CIFAR100 and Small LSUN).

**Strengths:**

1.Three classification benchmarks.
2.Two backbones for experiments.

**Weaknesses:**

1.This paper is not well organized, both writing and format, and the main paper spans only 7 pages, leaving much of the experimental detail to the appendix.
2.The paper mainly provides an empirical comparison and does not introduce a novel algorithmic contribution or conceptual framework.
3.The paper claims 20 experimental repetitions per setup, but no error bars, standard deviations, or significance markers are shown in the figures.
4.Although the paper claims full code availability (Section 8), the provided repository link on anonymous.4open.science is not available.
5.Task domain in experiment is limited to computer vision.

**Questions:**

See Weaknesses.

---

> ### Author Response · Authors · 2025-11-24
>
> We thank the reviewer for their review.
> Even though, we will not be submitting an updated PDF to this rebuttal, we would like to distance ourselves from some of the addressed weaknesses:
>
> 1. Only 7 pages: In our opinion, the length of a paper should in no way, shape or form influence the acceptance decision. Missing important information in the paper is valid, but from your comment, all the information was present (if only in the appendix)
> 2. No novel method or framework: We deliberately have not introduced novel approaches in this work, as this would confound our findings. We mainly tried to generate additional insight about the "state" of active learning, trying to uncover why so many algorithms in AL do not stand the test of time
> 3. No error bars, or standard deviations: Base most of our reporting on Critical-Difference diagrams, which evaluate statistical significance with the Wilcoxon signed rank test and display respective markers. This makes error bars or standard deviations entire obsolete
>
> Points 4. and 5. are valid: \
> We missed a problem with the provided link.\
> Also, adding more domains to this work should always be the goal.\

---

### Official Review · Reviewer_kzzo · 2025-11-01

**Soundness:** 1
**Presentation:** 1
**Contribution:** 2
**Rating:** 2
**Confidence:** 4

**Summary:**

The paper benchmarks AL against data augmentation and semi/self-supervised learning under a pool-based setting with a fixed labeling budget. Experiments use CIFAR10, CIFAR100 and a small LSUN subset with ResNet-18/EfficientNet-V2; each acquisition round adds a fixed batch of labeled points, and performance is summarized by the AUC over rounds. when AL is combined with strong DA and SSL techniques, it surprisingly is still able to provide improvements. Based on these results, the paper frame AL not as a method to combat missing labels, but as the final building block to squeeze the last bits of performance out of data after appropriate DA and SSL methods as been applied.

**Strengths:**

The question mentioned in the paper is important, "Is AL still worthwhile under modern training pipelines with limited labels?”

**Weaknesses:**

1. The title/framing (“role of active learning in modern deep learning”) suggests broad conclusions, yet evidence is confined to small. Modern deep learning goes far beyond the experiment the paper conducted. In contrast, the paper scale image classification with limited DA/SSL variants and non-modern backbones. The paper should at least provides experiments on ViT/DeiT. Claims that generalize to “modern pipelines” or provide prescriptive guidance across domains (larger images, detection/segmentation, NLP, multimodal, distribution shift, class imbalance) are not supported by the presented scope.
2. The paper obtains Insufficient workload as well as limited contribution. The bulk of the effort is engineering rather than advancing methods or analysis. Only small/medium image classification datasets (CIFAR/Small-LSUN) and two CNN-style backbones; no ViT/DeiT or larger-scale settings.
3. Substantial compute was spent “running baselines,” but the paper adds little in terms of new algorithmic ideas, theoretical insight, or explanatory analysis.
4. Almost all the references to the pictures and appendices in the text have no attached links, which is not convenient for readers to understand this article.

**Questions:**

1. Can you add ViT/DeiT backbones and at least one larger-scale or distribution-shift/long-tail setting to justify claims about “modern deep learning”?
2. Include more modern DA/SSL (e.g., RandAugment/CutMix/TrivialAugment, DINO/MAE/DINOv2) to ensure conclusions aren’t an artifact of conservative choices.
3. Report the impact of acquisition batch size and total label budget on both performance and compute; do method rankings change?
Reduce overclaim: Reframe conclusions to explicitly state the narrow domain where they are supported, or provide new experiments/diagnostics (e.g., uncertainty calibration, representation analyses) to substantiate broader claims.

---

> ### Author Response · Authors · 2025-11-24
>
> We thank the reviewer for their thorough work. \
> We will not be submitting an updated PDF to this rebuttal.
>
> Unfortunately, extending our research to larger datasets and models was not possible, due to computational constraints.
> However, we acknowledge the miss-match between the title and the provided experiments.

---

### Official Review · Reviewer_tt6Z · 2025-11-07

**Soundness:** 1
**Presentation:** 3
**Contribution:** 1
**Rating:** 0
**Confidence:** 5

**Summary:**

On this work, the authors do an empirical study in finding the advantages/usability of active learning, and comparing it with using instead data augmentation and semi-supervised learning. The main premise of the paper, is that compared to the two other approaches, active learning does not give significant performance boost, while at the same time not improving the performance when combined with the other two techniques. They validate this in the experimental section.

**Strengths:**

Readability - While short, the paper is well-written and very easy to understand. Both the diagrams and the figures support the writing of the paper.

**Weaknesses:**

Unfortunately, I think the paper has several big weaknesses, that make it not suitable for publication:

1) In some ways, the paper posits the story as active learning, data augmentation and semi-supervised learning are distinct approaches, with the models using one of them. This is fundamentally wrong. Pretty much any active learning method is always combined with a data augmentation method both in practice and in research papers. Data augmentation is also inherently a part of semi-supervised learning (see all consistency methods, or teacher-student models that use strong/weak augmentations). Furthermore, several methods combine active learning with semi-supervised learning showing that the final performance significantly improves over both strategies in isolation. So

[A] Gao et al., Consistency-based semisupervised active learning: Towards minimizing labeling cost, ECCV 2020 (270 citations).
[B] Yu et al., Consistency-based Active Learning for Object Detection, CVPRw 2022 (85 citations)
[C] Elezi et al., Not All Labels Are Equal: Rationalizing The Labeling Costs for Training Object Detection, CVPR 2022 (64 citations).

These works are neither recent, nor fringe, considering that they have been published in top-tier venues several years ago and are quite cited). They all show the same thing: (1) Active learning in isolation does not perform as well as semi-supervised learning; (2) They all are trained together with some data augmentation strategy; (3) They show how active learning combined with semi-supervised learning improves over either in isolation. (1) and (2) make this paper's claims redundant, while (3) makes them wrong.

From my personal experience, using active learning in both research and industry, active learning improves when combined with semi-supervised learning.

2) The datasets used for experiments are outdated and small. At the very least, the authors should have used ImageNet for classification (which nowadays is also quite a small and outdated dataset). I do not think modern papers can be evaluated in CIFAR datasets when it comes to top-tier publications.

3) From the paper title and abstract, this is an active learning paper. However, everything is set in the context of classification. This should have been clarified in the title and abstract, that it is about active learning in classification. Or ideally, show experiments in other domains such as at the very least detection, but also segmentation, tracking, etc if it is meant to be a general active learning paper.

4) The title is about active learning in modern deep learning. But, the experiments are mostly in using small CNNs that are a decade ago such as ResNets. There are no Transformers in the paper. In fact, from the title, I was quite excited and was expecting active learning in the context of LLMs or VLMs, in what is basically modern deep learning.

**Questions:**

I do not have any questions, but I do not think the paper is suitable for a top-tier publication. The claims are wrong, the experiments are limited in small datasets and CNNs.

---

> ### Author Response · Authors · 2025-11-24
>
> We thank the reviewer for their comprehensive review.
> We will not be submitting an updated PDF to this rebuttal.
>
> Your raised concerns are all valid.
> Especially, missing baselines A-C is not acceptable. We will be working to incorporate these publications into our research.

---

### Official Review · Reviewer_kBGg · 2025-11-09

**Soundness:** 2
**Presentation:** 3
**Contribution:** 2
**Rating:** 4
**Confidence:** 5

**Summary:**

This paper studies how effective active learning really is when compared with other methods for dealing with limited labeled data. The authors explore three main techniques, data augmentation (DA), semi-supervised learning (SSL), and active learning (AL), in realistic small-label settings. They find that while DA and SSL can together improve model performance by up to 60%, AL only adds a small 1–4% gain over random sampling. However, when combined with strong DA and SSL pipelines, AL can still provide marginal improvements.

**Strengths:**

This paper offers a comprehensive, well-designed study of how AL performs in modern deep learning settings that already use DA and SSL. Its originality lies in shifting the focus from proposing new algorithms to questioning the practical value of AL under realistic conditions. The experimental setup is solid and carefully controlled across datasets and models, which makes the conclusions credible. The results are clearly presented and give an honest, data-driven picture: while DA and SSL deliver large gains, AL offers only a small improvement.

**Weaknesses:**

1. Although the paper aims to evaluate the practical usefulness of AL, its experimental design assumes that we can first fully tune and identify the best DA and SSL combination for each dataset before applying AL. This actually contradicts the real motivation for AL, which is to operate in scenarios with limited labels and limited resources, where such exhaustive tuning is infeasible. In practice, DA, SSL, and AL are usually co-designed and co-adapted, not applied sequentially in a pipeline where DA/SSL are fully optimized first.
2. According to the experimental figures (e.g., Figure 3, Figure 13, Figure 25), the starting points across methods are not aligned, which suggests that the 20 repeated runs do not use paired random seeds and therefore each method begins with a different initial labeled pool. Since no acquisition occurs in round 0, all methods should theoretically start from identical performance when DA, SSL, and training protocols are fixed. More importantly, the paper reports AL gains of only 1–4%, which is comparable to the expected fluctuation caused by different initial labeled pools. Without controlling for seed alignment, it is unclear whether these small gains truly reflect the effect of AL or merely dataset-level randomness.
3. Some AL strategies evaluated in the paper, such as BALD, Margin Sampling, BADGE, and CoreSet, rely on specific assumptions about the geometry of the feature space (e.g., uncertainty calibration, margin smoothness, linear separability, gradient embedding structure). However, strong SSL methods like SimCLR are known to significantly reshape the representation space by increasing local density, smoothing decision boundaries, and altering cluster structure. These shifts can directly affect whether AL’s underlying assumptions still hold, and the paper does not examine how SSL transforms the feature space or how such transformations influence AL behavior.

**Questions:**

1. This is related to Weakness 3, embedding visualizations should be added, without it, it is unclear whether the AL methods are being evaluated in conditions where they remain theoretically appropriate.
2. This is related to Weakness 1, since all experiments are repeated 20 times, adding error bars (or shaded confidence intervals) to the accuracy curves would make it easier to judge whether the differences between methods, especially the small 1–4% AL gains, would be more meaningful.
3. The current benchmarks (CIFAR, SVHN, FashionMNIST) are small, clean, and balanced, where strong SSL methods are known to perform extremely well. This leaves very little room for AL to demonstrate meaningful gains. Have the authors considered evaluating AL under more challenging datasets?

---

> ### Author Response · Authors · 2025-11-18
>
> We thank the reviewer for their review of our work.
>
> Even though, we will not be submitting a rebuttal, we would like to address some of the raised concerns:
>
> Weakness 1: This is a very valid point, as we failed to properly explain our reasoning behind this unobtainable tuning of DA and SSL methods for each dataset: In our opinion, stronger DA and SSL methods provide a more "saturated" training pipeline in which it is harder to achieve further lifts. So, if AL is able to show a lift in the "hardest" (most optimized) environment, we can extrapolate this result to many real world scenarios with less optimized DA and SSL.
>
> Weakness 3: We actually believe, that the geometry of the feature space does not play a big role in algorithms like Coreset or Badge, as both techniques anyways operate on the encoded versions of the inputs (i.e. they are calculated on the representations of the penultimate layer of the current model). In that regard, nothing changes when we apply pre-trained weights to the model.
>
> Question 3: This an oversight by us. Thank you for the hint, we will be incorporating less clean datasets into our experiments.

---

### Official Review · Reviewer_xioc · 2025-11-12

**Soundness:** 1
**Presentation:** 1
**Contribution:** 1
**Rating:** 2
**Confidence:** 4

**Summary:**

This paper investigates whether active learning (AL) is still useful when combined with modern techniques like data augmentation (DA) and semi-supervised learning (SSL). Their conclusion: Active learning is the least efficient way to handle low-data regimes and should only be used after applying DA and SSL to squeeze out the last few percentage points of performance. They call for a paradigm shift, arguing that developing or evaluating AL methods without modern DA and SSL is no longer scientifically valid.

**Strengths:**

The authors evaluate a broad set of AL algorithms (BADGE, Galaxy, CoreGCN, uncertainty sampling variants) across multiple datasets and architectures (ResNet18, EfficientNetV2), under consistent, reproducible conditions. They incorporate DA and SSL in a structured, layered fashion and repeat all experiments 20 times — an almost unheard-of level of rigor in this domain.

**Weaknesses:**

* The paper sometimes feels like a manifesto disguised as a benchmark. Its stance - that AL research should stop ignoring DA/SSL - is correct but somewhat overstated. There might still exist niches where AL’s cost-benefit calculus makes sense (e.g., expensive medical labeling), and the paper’s rhetoric risks alienating parts of the community that could instead build on its insights.
* The combination of Active Learning and Semi-Supervised Learning has already been explored in prior literature. The authors should better clarify how their work differs from, or extends beyond, existing studies in this area.
* The figures are poorly crafted.
* This paper only has 7 pages.
* This paper seems not finished yet.
* Another limitation is that the study remains vision-centric.

**Questions:**

see above

---

> ### Author Response · Authors · 2025-11-18
>
> We thank the reviewer for their honest criticism of our work.
>
> Even though, we will not be submitting a rebuttal, we would like to distance ourselves from the statement in the first point of your "Weakness" category.
> This point seems to suggest, that this work should not be accepted, simply because it might make research in AL (e.g. for medical applications) less attractive.
> We are of the opinion, that our work is important for future research in AL, as most of our advancements in the past decade have not withstood the test of time and are rarely applied in practice and this paper can provide a piece of evidence about the reasons for this.

---

### Meta-Review · Area_Chair_h8ku · 2026-01-12

**Summary:**

This paper has been reviewed by five reviewers, and all the reviewers have unanimously agreed that this paper does not meet the standards of acceptance at ICLR. The authors have not submitted a rebuttal, and in their responses have agreed on several issues raised. I would encourage the authors to consider all the improvements suggested and resubmit to the next venue once all issues have been fixed.

**Reviewer Concerns:**

None of the concerns have been adressed.

**Reviewer Scores:**

None

---

### Decision · Program_Chairs · 2026-01-26

Reject